# Dietary Intake, Nutritional Status and Sensory Profile in Children with Autism Spectrum Disorder and Typical Development

**DOI:** 10.3390/nu14102155

**Published:** 2022-05-22

**Authors:** Paula Mendive Dubourdieu, Marcela Guerendiain

**Affiliations:** 1Departamento de Educación, Escuela de Nutrición, Universidad de la República, Montevideo CP 11600, Uruguay; 2Área de Investigación, Escuela de Nutrición, Universidad de la República, Montevideo CP 11600, Uruguay; mguerendiain@nutricion.edu.uy

**Keywords:** autism, food intake, nutritional status, sensory profile, gluten-casein free diet, food selectivity

## Abstract

Children with autism spectrum disorder (ASD) may consume a restricted diet, whether due to sensory sensitivities or an adherence to a gluten and casein free (GCF) diet. Our objective was to analyze dietary intake, nutritional status, and sensory profile in children with and without ASD. A descriptive, cross-sectional study was carried out in 65 children (3–12 years, ASD = 35, typical development (TD) = 30). Short Sensory Profile and food frequency questionnaires were applied. All participants were categorized into normal weight and excess weight, typical sensory performance (TP), and probable + definite difference (PD + DD); and ASD group into GCF dieters (ASD-diet) and non-dieters (ASD-no diet). Children with ASD had a higher intake (gr or ml/d) of vegetable drinks (*p* = 0.001), gluten-free cereals (*p* = 0.003), and a lower intake of fish (*p* < 0.001) than TD ones. The ASD group showed a lower score in total sensory profile score (*p* < 0.001) than TD group. In the ASD group, those who had PD + DD in their sensory profile consumed fewer dairies (*p* = 0.019), and more cereals (*p* = 0.036) and protein foods (*p* = 0.034) than those with TP. These findings confirm the need to consider the neurodevelopment, sensory profile, and type of diet to improve the ASD child’s nutrition. Further long-term research is needed to explore their impact on health.

## 1. Introduction

Worldwide prevalence of people diagnosed with autism spectrum disorder (ASD) is increasing and, in 2019, the US Centers for Disease Control and Prevention (CDC) estimated that 1 in 59 children had ASD [1,2]. Multiple environmental, immunologic, and genetic factors play a role in its pathogenesis [3]. Hence, interest in the effect of special diets and nutrition on autism is increasing, particularly as a way to improve behavior, attention span, social interaction, and eye contact [4]. Many studies have shown that some children and adolescents with autism are on a gluten and casein free (GFCF) diet [5,6]. Graf-Myles et al. reported that diet restrictions can impact dietary intake of cereals and dairy and may also lead to a lower intake of calcium and grains supplemented with folate in ASD group as compared to TD group [7]. 

Furthermore, rigid and repetitive dietary patterns are frequently observed in this population [1]. Some sensory processing problems, such as sensory modulation expressed as hyper and/or hyposensitivity, seem to make it more challenging for a child to adapt to new foods and have an impact on their development [2]. On the other hand, neurotypical children around the age of six often show a preference for certain foods and a rejection of others as a part of their developmental age [3,4]. Many influencing factors can affect an individual’s food choices, and studies suggest that eating disorders in autism may be one significant contributor to comorbidities such as gastrointestinal symptoms [2,5,6]. Additionally, some studies show that children with ASD have greater rates of overweight or obesity than typically developing (TD) ones, and this fact could be related to unusual dietary patterns and decreased opportunities for physical activity [7]. Furthermore, a normal body mass index (BMI) might hide nutritional inadequacies. A rejection of the intake of certain food groups, such as those rich in protein, and an increased consumption of caloric high-fat foods have been observed in children with ASD [8]. 

Although children and adolescents with autism are known to be highly selective when choosing food and tend to pick specific food textures, colors, smells, or other characteristics, only a few studies currently exist examining their food choices or comparing them to TD children [4]. Much available and public research focuses on nutrients or supplement intake but does not provide information on their diet or nutritional status. Therefore, our objective was to describe and analyze children’s nutritional status with anthropometric measures, dietary intake, and sensory profile using the Short Sensory Profile (SSP) parent-reported questionnaire in children and adolescents with ASD and TD [9]. 

## 2. Materials and Methods

### 2.1. Participants and Study Design

A descriptive, cross-sectional study was carried out in 65 children aged 3 to 12 years with ASD (n = 35) and TD (n = 30), recruited in Montevideo, Uruguay; participants in research project *Alimentación, nutrición y salud intestinal en niños y adolescentes con Trastorno del Espectro Autista y neurotípicos* (food, nutrition, and intestinal health in children and adolescents with autism spectrum disorder and neurotypicals). Participants were recruited through advertising the study in parents of children with autism organizations and at autism therapy centers. Inclusion criteria for the ADS group was a clinical diagnosis by a psychiatrist or a pediatric neurology specialist, confirmed by *Diagnostic and Statistical Manual of Mental Disorders, Fifth Edition* (DSM-V) criteria [10]. The TD group included children with no neurodevelopmental alterations. Those diagnosed with attention deficit and hyperactivity disorder, diabetes mellitus, genetic diseases, inborn errors of metabolism, inflammatory bowel disease, celiac disease, motor disability, or without informed parental consent were excluded from both groups. No children in the TD group were on a restricted diet.

The study meets all ethical requirements for human studies stated in the Helsinki Declaration 2000 and established in Uruguayan regulations. It was approved by the Research Ethic Committee of the School of Nutrition, Universidad de la República, and registered with the Ministry of Health of Uruguay (no. 282599).

### 2.2. Anthropometric Measures

Anthropometric variables were assessed during an interview with the children and their parents, and an informed consent form was previously signed. For anthropometric measurements of weight and height, participants wore light clothing and were barefoot according to techniques standardized by Frisancho and the World Health Organization (WHO) [11,12]. Measurements were performed by the same nutritionist researcher in triplicate in order to avoid interobserver errors and were later averaged. Weight was measured using a portable electronic scale (Seca 813, Hamburg, Germany), with a maximum capacity of 200 kg and an accuracy of 100 g. Height was measured using a portable height rod (208 Seca) with a range of 810 to 2060 mm and a precision of 1 mm. Readings were recorded in meters and centimeters. Birth weight data were taken from the pediatric card of each participant.

Nutritional status was assessed according to the height-for-age (H/A) and body mass index-for-age (BMI/A) indicators, expressed in z-score (z). Software Anthro (for children aged 3 to 5 years) and Anthro plus (for children and adolescents aged 5 to 12 years) (WHO v.1.0.2, 2007), which apply WHO child growth curves [12], were used. Cut-off points used for the BMI/A of children aged 2–5 years are: >3SD, obesity; >2SD, overweight; >1SD, risk of overweight; between <1SD and >−1SD, normal weight; ≤−1SD, risk of wasting; ≤−2SD, emaciation; ≤−3SD, severe emaciation. In those over 5 years old: ≥2SD, obesity; ≥1SD, overweight; between <1SD and >−2SD, normal weight; ≤−2SD, wasting, ≤−3SD; severe emaciation. For the purpose of analysis, both age groups were unified into two categories: normal weight (NW) and excess weight (EW) (risk of overweight + overweight + obesity). The use of cut-off points for deficit malnutrition (risk of wasting, wasting and severe wasting) was ruled out since sample size was small (ASD *n* = 4; TD *n* = 1); therefore, these participants were not considered for our anthropometric analysis. 

### 2.3. Dietary Intake

Children with ASD were classified into two groups, one consisting of those on a gluten-free and casein-free diet (ASD-diet, *n* = 19), and a second one formed by those without a restricted diet (ASD-no diet, *n* = 16).

During the same interview where anthropometric data were taken, information on the dietary intake over the past 3 months was collected by the nutritionist researcher. The SAYCARE study food frequency questionnaire (FFQ) [13], validated for children and adolescents from seven cities in Latin America, was applied. It includes a photographic atlas with the weight of each food. This questionnaire was adapted to our subject population in order to obtain further information on the consumption of gluten-free and casein-free foods, due to its relevance to our study. A food photo booklet was shown to caregivers and children with the FFQ for them to identify food portion size. The amount of food consumed was described using home measures and then converted into grams or ml, depending on the type of food. The daily intake of each food was estimated taking consumption frequency into account.

Foods were organized into different groups as follows: (1) dairy products, ‘total dairy’: milk, yogurt, chocolate milk, dairy desserts, cheese; (2) ‘vegetable drinks’: birdseed, chestnut, almond, oat, rice, and coconut drinks; (3) cereals, ‘cereals with gluten’: pasta, bread, cookies, bakery products, breakfast cereals, pizza, and empanadas (dough stuffed with meat, fish, vegetables, etc. baked or fried), ‘cereals without gluten’: the same foods in the previous group without gluten, and rice; (4) meats and derivatives, and eggs, ‘total protein foods’: meat, minced meat, chicken, pork, eggs, fresh and canned fish and *milanesa* steak with and without gluten (a thin slice of beef dipped in beaten eggs and breaded; the fact that 25% of its weight is due to cereal has been taken into account); (5) ‘total high-fat foods’: butter, ghee (fat obtained by heating cow milk butter), and oils.

### 2.4. Sensory Sensitivity 

Parents completed a validated online questionnaire to determine their children’s sensory features. The Spanish version of the Short Sensory Profile (McIntosh et al. 1999) questionnaire was applied to establish the frequency of a child’s sensory, behavioral, or emotional responses to daily life events. The questionnaire contains seven subscales: tactile sensitivity, taste/smell sensitivity, movement sensitivity, under responsive/seek sensation, auditory filtering, low energy/weak, and visual/auditory sensitivity. Each item represents observable child behaviors and is rated on a five-point scale ranging from ‘always’ to ‘never’ (1: always, 2: frequently, 3: occasionally, 4: rarely, 5: never), resulting in a potential maximum score of 190. The total score obtained can be classified into three categories: typical performance (TP: 190–155), probable difference (PD: 154–142), definite difference (DD: 141–38). In our case, due to a small sample size, we grouped PD and DD together, and therefore two categories were used (TP and PD + DD).

#### Statistical Analysis

IBM SPSS Statistics 22.0 (IBM Corp, Armonk, NY, USA) was used for statistical analyses. Results were expressed as means ± standard deviation (SD), for quantitative variables. The Kolmogorov–Smirnov test was used to assess variable distribution. Independent sample *t*-test (for parameters with normal distribution) and Mann–Whitney test (variables without normal distribution) was carried out to analyze anthropometric characteristics, dietary intake, and sensory profile score, according to neurodevelopment (ASD or TD) and autistic children’s diet (ASD-d or ASD-nd). A *p*-value < 0.05 was set for statistical significance (two-tailed).

To study dietary intake according to nutritional status (NW and EW) or sensory profile score (TP and PD + DD) in ASD and TD, foods without a normal distribution were log transformed (milk + yogurt, cheese, total dairy, cereals with and without gluten, meat, minced meat, chicken, pork, *milanesa* with and without gluten, eggs, fish, total protein food, butter, ghee, oils, and total food source of fat). Variable analysis was performed applying two-way ANCOVA (adjusted for birth weight, total sensory profile score and with GFCF diet/not restricted diet; or birth weight, nutritional status and with GFCF diet/not restricted diet; respectively), exploring possible main effects of factors and interactions among them. Pair comparisons between the different groups were adjusted by Bonferroni post hoc test. Comparisons between NW and EW or between TP and PD + DD in all participants (‘All’) were carried out using one-way ANCOVA and correcting for potential confounders (birth weight, total sensory profile score, and with GFCF diet/not restricted diet; or birth weight, nutritional status, and with GFCF diet/not restricted diet, respectively).

## 3. Results

Children’s anthropometric characteristics and dietary intake are presented in Table 1. Mean height, weight, BMI, BMI-for-age Z-score, height-for-age z-score, and birth weight showed no significant differences between ASD-diet and ASD-no diet groups, and between all children with autism (ASD-t) and the TD group. However, there is a significant difference in mean age between ASD-t group and TD group (*p* = 0.045).

As expected, children in the ASD-diet group consumed a significantly lower amount of total dairy products (*p* < 0.001), cereals with gluten (*p* < 0.001), total cereals (*p* = 0.005), *milanesa* with gluten (*p* < 0.001), and butter (*p* = 0.016), than the ASD-no diet. The ASD-t group consumed a significantly lower amount of these foods than the TD group, total dairy products (*p* < 0.001), cereals with gluten (*p* < 0.001), total cereals (*p* = 0.009), *milanesa* with gluten (*p* = 0.014), and butter (*p* = 0.004). By contrast, the mean intake of vegetable drinks (*p* = 0.003), cereals without gluten (*p* = 0.001), *milanesa* without gluten (*p* = 0.004), and coconut oil (*p* < 0.001) was higher in ASD-diet than in the ASD-no diet group, and there was also a significant difference between the ASD-t group and neurotypicals (vegetable drinks: *p* = 0.001; cereals without gluten: *p* = 0.003; *milanesa* without gluten: *p* = 0.032; coconut oil: *p* = 0.006). Children with ASD who were on a GFCF diet consumed significantly more ghee (*p* < 0.008) and total foods source of fat (*p* = 0.007) than those without a restricted diet. Furthermore, in the ASD-t group fish intake was lower than in TD ones (*p* < 0.001).

Mean scores in the ASD-diet, ASD-no diet, ASD-t, and TD groups, for the different items of the SSP questionnaire, are displayed in Table 2. Scores for tactile sensitivity (*p* < 0.001), taste/smell sensitivity (*p* = 0.005), under responsive/seeks sensation (*p* < 0.001), auditory filtering (*p* < 0.001), and total sensory profile (*p* < 0.001) were significantly higher in TD group than in ASD-t group, showing a greater sensitivity in children with ASD. When diet was considered, the ASD-no diet group obtained a significantly higher score in visual/auditory sensitivity (*p* < 0.001) and low energy/weak (*p* = 0.010) than ASD-diet group.

In Table 3, the comparison of dietary intake between nutritional status (NW vs. EW) and neurodevelopment (ASD vs. TD) groups was analyzed by two-way ANCOVA to explore potential main effects of factors and interactions among them. Only the coconut oil intake showed a significant interaction between nutritional status and ASD/TD groups (F (1, 16) = 7.589, *p* = 0.014, ƞ^2^ = 0.322). The nutritional status factor also showed a significant main effect on coconut oil (F (1, 16) = 6.004, *p* = 0.026, ƞ^2^ = 0.273). Having ASD or TD had a statistically significant main effect on the intake of milk + yogurt (F (1, 30) = 6.299, *p* = 0.018, ƞ2 = 0.174). Children with a normal weight in ASD had a significantly lower mean intake of milk + yogurt (*p* = 0.007) and total dairy (*p* = 0.014), and a significantly higher mean intake of minced meat (*p* = 0.010) and coconut oil g/d (*p* = 0.011) than those with a normal weight in the TD group. Additionally, in the TD group, children with a normal weight consumed a significantly lower amount of coconut oil (*p* = 0.003) than those with excess weight. However, children in the group All (ASD + TD) with a normal weight had a higher intake of coconut oil (*p* = 0.026) than the excess weight ones.

In Table 4, the comparison of dietary intake between sensory profile (TP vs. P + DD) and neurodevelopment (ASD vs. TD) groups was studied by two-way ANCOVA. The sensory profile factor showed a significant main effect on cheese intake (F (1, 35) = 7.533, *p* = 0.009, ƞ^2^ = 0.177). The neurodevelopment factor showed a significant main effect on coconut oil (F (1, 17) = 4.489, *p* = 0.049, ƞ^2^ = 0.209). In the ASD group, children with TP consumed higher milk + yogurt (*p* = 0.009), total dairy products (*p* = 0.019), meat (*p* = 0.047); lower total cereals (with and without gluten) (*p* = 0.036); and total protein foods (*p* = 0.034) than PD + DD ones. Children with ASD and a TP had a lower intake of milk + yogurt (*p* = 0.004) and total dairy products (*p* = 0.037), and higher consumption of coconut oil (*p* = 0.049) than the TD children with TP. When all children (ASD + TD) were studied, those with TP showed a higher intake of cheese than the PD + DD group (*p* = 0.009).

## 4. Discussion

The aim of this study was to describe and analyze dietary intake in children with and without autism, in relation to their nutritional status and sensory sensitivity. Our main findings were that children with ASD who presented a higher sensory sensitivity (probable or definite difference in their sensory profile) had a lower intake of total dairy products and a higher intake of total cereals and protein foods than TP children with ASD.

Considering food intake exclusively, in agreement with our results after a systematic review of cohort studies, Ristory et al. showed that there was no difference in the intake of protein foods between ASD and TD groups [3]. However, regarding fish intake, we observed that children with ASD consumed a significantly lower amount of fresh and canned fish than TD group. Notably, another study analyzing food consumption and eating behavior in autistic children and their typically developing peers found that 27% of children with autism never ate fish [1]. This lower consumption may be worrisome because a higher fish intake has been associated with better cognitive function in adolescents in a Dutch study [14]. Likewise, Plaza-Díaz et al. outlined that low docosahexaenoic acid (DHA) levels have been associated with damaged language and motor skills in children, but they found not significantly differences in the average daily intake of the fatty acids eicosapentaenoic (EPA) + (DHA) contained in fish between the ASD and control group, perhaps due to a traditionally higher consumption of fish in Spain than in other counties [15]. 

Furthermore, even though there is no conclusive evidence proving an improvement of ASD symptom management in children on the GFCF diet, it is currently one of the most popular interventions [16]. A gluten-free diet consists in the exclusion of all foods containing wheat, barley, or rye as well as all flours, bread, pasta, and other bakery products made from these cereals; while a casein-free diet is based on avoiding dairy products such as milk, yogurt, cheese, butter, cream, or ice cream, among others [17]. As expected, we observed a lower consumption of dairy products and cereals with gluten in those autistic children whose were on the GFCF diet, and a lower intake of gluten and casein in the ASD total group (ASD-d + ASD-no diet) in comparison with TD participants. 

Consequently, a higher intake of vegetable drinks in the ASD group on a GFCF diet than among non-dieters may be due to the fact that they used such drinks as a substitute for dairy [18,19]. It is still unclear whether vegetable drink consumption may be associated with any beneficial effects on health [18]. These beverages are liquid-based extracts of nuts, oilseeds, legumes, cereals, and pseudo-cereals that simulate cow milk appearance and consistency but are known to have different nutrient properties and bioavailability [20]. Some of the industrialized beverages could be calcium-fortified, but as compared to milk, the nutritional profile of vegetable substitutes is generally low in proteins, vitamins (B2, D, E), mineral content (especially calcium), and saturated fat acid levels (except for coconut milk); and regarding their content of carbohydrates, they show good energy levels, similar to whole milk [18]. 

In line with our results, analyses involving nutritional data suggested children with ASD had a significantly lower consumption of calcium and vitamin D compared to their TD peers with bad consequences in bone and skeletal health [21,22]. In a cross-sectional study with 738 ASD children and 302 typically developing children recruited in China, those with ASD consumed less milk and fewer dairy products than TD ones [23]. In our research, we analyzed this characteristic also discriminating by nutritional status. Children with a normal weight and ASD showed a lower consumption of milk and yogurt and total dairy products, in comparison with those with a normal weight in the TD group, which was mainly driven by the ASD group dieters. It is worth noting that a low consumption, or even the elimination, of milk and other dairy products could result in calcium deficiency due to an intake below recommended levels [24]. 

As far as fat consumption is concerned, we found that ASD-diet had a higher intake of ghee and total fat intake than the ASD-not diet group. A higher consumption of ghee might be the result of it being considered a casein-free food derivate of butter and one of the options of a GFCF diet [25]. In addition, ghee is commonly used in Indian and Pakistani cooking, and has played an integral role as a homeopathic treatment for learning and memory enhancement [26]. However, there is a lack of scientific data to support this claim. Moreover, the intake of coconut oil showed a significant interaction between the nutritional status groups and the neurodevelopment factor. We observed a higher intake of coconut oil in those with a normal weight and ASD in comparison with TD with a normal weight. This could be explained by the fact that the ASD-diet group consumed a high amount of coconut oil. The use of this fat for treating epilepsy with ketogenic diets might be a reason for parents of children on an ASD-diet to consider it as beneficial for this common comorbidity [26,27]. However, the same randomized controlled trials show that coconut oil intake increases low-density lipoprotein cholesterol and total cholesterol when compared with other vegetable oils, and the current recommendation is to include it only within 10% of total caloric intake [28]. Interestingly, children with normal nutritional status in the TD group consumed significantly less coconut oil than the excess weight group, which is consistent with a broad range of studies proving a link between consumption of saturated fatty acids and an increase in fat cell (adipocyte) size and number [29]. 

Regarding the nutritional status, we found no significant difference of participants with ASD and TD. Nevertheless, in a meta-analysis of epidemiological studies examining the association between obesity, overweight, and ASD, the prevalence of obesity was significantly higher in individuals with ASD than in controls [27]. According to the authors, this could be related to eating disorders, such as food selectivity and more time spent on sedentary activities [27]. Its discrepancy with our results may be due to the small sample size of the population we studied.

Additionally, some studies report that ASD children have strong food preferences, and alterations in sensory processing—especially in tactile, taste, and smell sensitivity—may have an impact on health and nutritional status [3]. Coincidentally, upon applying the SSP questionnaire, we found that the participants with ASD had a lower score in ‘taste/smell sensitivity’ than TD participants, which points to a higher sensitivity in children with autism. The ASD group also showed higher tactile sensitivity, under responsive/seeks sensation, auditory filtering, and total sensory profile scores. A study using the same questionnaire for 281 children with ASD compared to age-matched peers with typical development reported similar results, showing ASD children´s greater sensitivity in the same aspects [28]. 

Sensory sensitivity has been suggested as a possible mechanism to explain food selectivity in children with ASD [4]. Among all participants (ASD + TD), those who have PD + DD had a lower mean intake of cheese than TP group, and the sensory profile factor had a significant main effect on cheese intake. On this topic, we observed that children with ASD and PD + DD (more sensitivity), consumed lower milk and yogurt, and fewer total dairy products, and higher total cereals with and without gluten, and protein foods than TP children with ASD. No difference was found in the short sensory profile between dieters and non-dieters, but as was expected ASD-t group had a higher sensory sensitivity than TD. Furthermore, this sensitivity was related to a lower dairy consumption and a higher intake of protein foods and cereals; however, Zimmer et al. described a lower intake of proteins in selective eaters with ASD [21]. Therefore, sensitivity could be considered an influencing factor in the consumption of these foods. 

Nonetheless, we also have found a greater consumption of non-traditional foods—such as different vegetable drinks, ghee, or coconut oil—in children on a GFCF diet. Therefore, it is important to determine their consumption and characteristics in order to assess diet quality for optimal growth and development, and to provide a proper nutritional treatment and follow-up, considering factors that may affect their dietary intake, such as sensory sensitivity. It is important to have a good knowledge of the different diets that children and adolescents with ASD are following to continue to study them and if it covers the recommended nutritional intakes so that health workers can provide scientific information in this regard. This finding attests once more to the need for personalized dietary recommendations for ASD [27,28]. 

The limitations of this study were problems inherent to the use of FFQ, since it is not a specific tool designed to address food consumption in children with autism. Quantities of consumed food are often estimated by caregivers rather than accurately measured, and reporting by parents might be biased towards a healthier diet. In addition, the small sample size has limited the results of the study. However, as far as we know, there are no previous studies providing information on these children’s nutritional status, food intake, and sensory profile as compared to neurotypical ones. We therefore think that this study may be an important contribution to increase knowledge on these topics.

## 5. Conclusions

To conclude, this study found differences in the intake of some food groups as per the sensory sensitivity and nutritional status grouping criteria between ASD and TD children. Predictably, we observed higher sensitivity in the ASD group in comparison to TD ones, while nutritional status showed no significant difference between both groups. Our findings also show a dietary difference in the intake not only of total dairy and cereals (with or without gluten), but also of other foods—such as vegetable drinks, ghee, coconut oil, and total foods’ source of fat—between the ASD-diet and ASD-no diet groups, with higher consumption in the ASD-diet group. Additionally, there was a higher intake of total dairy, total cereals, fish, and butter, and a lower mean intake of coconut oil in TD ones than in ASD total group. Our results have therefore given us a deeper knowledge of food consumption in this population and could be linked to the causes of nutritional deficiencies presented in children with ASD, although further research with a larger sample size is needed to continue to investigate connections among food intake, nutritional status, and sensory sensitivity, find out if they meet the nutritional requirements, and understand the causes to carry out intervention strategies if required with important implications for child health.

## Figures and Tables

**Table 1 nutrients-14-02155-t001:** Anthropometric characteristics and dietary intake in children with autism spectrum disorder and typical development.

Parameters	ASD Groups			ASD-Total Group	TD Group	*p ***
	ASD-Diet	ASD-No Diet	*p **			
** *Anthropometric characteristics* **	(*n* = 19)	(*n* = 16)		(*n* = 35)	(*n* = 30)	
Age (years)	6.05 ± 2.27	5.57 ± 1.91	0.688 ^1^	5.83 ± 2.10	7.19 ± 2.56	**0.045 ^1^**
Height (cm)	118.11 ± 13.74	117.83 ± 12.83	0.882 ^2^	117.98 ± 13.14	125.43 ± 18.31	0.069 ^2^
Weight (Kg)	23.47 ± 6.25	24.73 ± 6.00	0.766 ^1^	24.05 ± 6.08	29.37 ± 11.78	0.266 ^1^
BMI (Kg/m^2^)	16.60 ± 2.26	17.71 ± 2.27	0.233 ^1^	17.11 ± 2.30	17.89 ± 2.70	0.469 ^1^
BMI for Age Z-score	0.50 ± 1.48	1.37 ± 1.32	0.078 ^1^	0.89 ± 1.45	0.92 ± 1.04	0.921 ^1^
Height for Age Z-score	−0.003 ± 1.17	0.58 ± 1.12	0.140 ^2^	0.26 ± 1.17	0.31 ± 1.19	0.833 ^2^
Birth Weight (Kg)	3179.37 ± 690.53	3519.69 ± 694.47	0.157 ^2^	3334.94 ± 703.42	3268.63 ± 462.58	0.651 ^2^
** *Dietary intake* **	(*n* = 18)	(*n* = 15)		(*n* = 33)	(*n* = 29)	
*Dairy products*						
Milk + Yogurt (g/day)	0.00 ± 0.00	191.89 ± 262.56	**<0.001 ^1^**	87.22 ± 198.94	353.40 ± 237.32	**<0.001 ^1^**
Cheese (g/day)	0.71 ± 2.41	15.62 ± 14.17	**<0.001 ^1^**	7.49 ± 12.15	16.60 ± 15.42	**0.002 ^1^**
T. Dairy (g/day)	0.71 ± 2.41	284.98 ± 275.95	**<0.001 ^1^**	129.92 ± 232.33	401.95 ± 243.96	**<0.001 ^1^**
*Vegetable drinks*						
Vegetable drinks (ml/day)	399.33 ± 415.82	42.26 ± 104.59	**0.003 ^1^**	237.03 ± 359.50	26.72 ± 118.38	**0.001 ^1^**
*Cereals*						
Cereals with gluten (g/day)	5.78 ± 10.32	210.32 ± 127.05	**<0.001 ^1^**	98.75 ± 133.47	236.58 ± 107.65	**<0.001 ^1^**
Cereals without gluten (g/day)	138.97 ± 88.08	42.61 ± 65.21	**0.001 ^1^**	95.17 ± 80.39	27.14 ± 57.71	**0.003 ^1^**
T. Cereals (with and without gluten) (g/day)	144.76 ± 88.37	252.93 ± 119.18	**0.005 ^2^**	193.93 ± 115.56	266.68 ± 104.90	**0.009 ^2^**
*Meats and derivatives, and eggs*						
Meat (g/day)	43.15 ± 26.20	34.05 ± 27.09	0.291 ^1^	39.01 ± 26.59	31.65 ± 18.20	0.335 ^1^
Minced meat (g/day)	50.70 ± 114.79	21.06 ± 23.91	0.340 ^1^	37.23 ± 86.46	19.49 ± 14.57	0.718 ^1^
Chicken (g/day)	34.38 ± 27.98	48.60 ± 58.34	0.360 ^1^	40.84 ± 44.24	29.51 ± 19.35	0.466 ^1^
Pork (g/day)	14.92 ± 22.21	3.21 ± 4.61	0.128 ^1^	9.60 ± 17.50	2.57 ± 3.59	0.307 ^1^
Milanesa with gluten (g/day)	3.07 ± 7.17	28.04 ± 45.10	**<0.001 ^1^**	14.42 ± 32.81	16.10 ± 12.74	**0.014 ^1^**
Milanesa without gluten (g/day)	10.25 ± 16.25	0.35 ± 1.35	**0.004 ^1^**	5.75 ± 12.89	1.24 ± 4.33	**0.032 ^1^**
Eggs (g/day)	43.70 ± 68.38	29.00 ± 34.70	0.456 ^1^	37.01 ± 55.37	27.37 ± 18.79	0.588 ^1^
*Fish*						
Fresh fish (g/day)	0.04 ± 0.18	0.00 ± 0.00	0.361 ^1^	0.02 ± 0.13	8.22 ± 7.99	**<0.001 ^1^**
Canned fish (g/day)	0.00 ± 0.00	0.00 ± 0.00	0.533 ^1^	0.00 ± 0.00	5.73 ± 9.89	**<0.001 ^1^**
Fish (fresh and canned) (g/day)	0.04 ± 0.18	0.00 ± 0.00	0.361 ^1^	0.02 ± 0.13	13.97 ± 12.54	**<0.001 ^1^**
T. foods source of proteins (g/day)	206.05 ± 129.84	183.21 ± 98.64	0.656 ^1^	195.67 ± 115.53	160.84 ± 50.69	0.386 ^1^
*Butter, ghee, and oils*						
Butter (g/day)	0.07 ± 0.33	2.17 ± 6.35	**0.016 ^1^**	1.03 ± 4.34	2.93 ± 5.36	**0.004 ^1^**
Ghee (g/day)	1.90 ± 5.85	0.00 ± 0.00	**0.008 ^1^**	1.03 ± 4.37	0.06 ± 0.26	0.201 ^1^
Coconut oil (g/day)	9.69 ± 11.51	0.48 ± 1.28	**<0.001 ^1^**	5.50 ± 9.63	0.95 ± 3.49	**0.006 ^1^**
Oil (g/day)	27.16 ± 18.69	18.60 ± 11.73	0.122 ^1^	23.27 ± 16.26	20.23 ± 13.24	0.494 ^1^
T. foods source of fat (g/day)	40.22 ± 22.93	21.25 ± 13.85	**0.007 ^1^**	31.60 ± 21.33	24.18 ± 15.57	0.203 ^1^

Results are expressed as mean ± SD. Statistically significant differences (indicated in bold): *p* < 0.05 (^1^ Mann–Whitney test, ^2^
*t* test). ASD: autism spectrum disorder, ASD-diet: ASD children with dietary intake without gluten and casein, ASD-no diet: ASD children without restricted diet, ASD-total: ASD children with and without restricted diet. TD: typical development, BMI: body mass index, FFQ: food frequency questionnaire, T: total. Comparison between: * ASD-diet and ASD-no diet, ** ASD-total group and TD group.

**Table 2 nutrients-14-02155-t002:** Sensory profile in children with autism spectrum disorder and typical development.

Sensory Profile Components (Score)	ASD-Diet (*n* = 19)	ASD-No Diet (*n* = 16)	*p **	ASD-Total (*n* = 35)	TD (*n* = 30)	*p ***
Tactile sensitivity	29.37 ± 3.23	29.19 ± 4.47	0.891 ^1^	29.29 ± 3.79	33.40 ± 1.69	**<0.001 ^1^**
Taste/smell sensitivity	14.21 ± 4.82	12.06 ± 5.77	0.239 ^1^	13.63 ± 2.27	16.83 ± 3.48	**0.005 ^1^**
Movement sensitivity	13.32 ± 2.79	14.00 ± 1.46	0.411 ^1^	13.87 ± 1.40	14.00 ±1.64	0.629 ^1^
Under responsive/seeks sensation	21.37 ± 5.18	21.13 ± 5.57	0.894 ^2^	21.26 ± 5.28	29.27 ± 4.88	**<0.001 ^2^**
Auditory filtering	20.79 ± 3.22	21.19 ± 3.92	0.744 ^1^	20.97 ± 3.51	27.23 ± 1.77	**<0.001 ^1^**
Low energy/weak	25.84 ± 4.91	29.06 ± 1.80	**0.010 ^1^**	27.31 ± 4.10	27.23 ± 3.31	0.546 ^1^
Visual/auditory sensitivity	18.16 ± 4.41	24.13 ± 1.31	**<0.001 ^1^**	20.89 ± 4.49	21.13 ± 4.16	0.841 ^1^
T. Short Sensory Profile	143.28 ± 16.35	150.75 ± 17.95	0.301 ^2^	146.70 ± 17.26	169.10 ± 10.39	**<0.001 ^2^**

Results are expressed as mean ± SD. Statistically significant differences (indicated in bold): *p* < 0.05 (^1^ Mann–Whitney test, ^2^
*t* test). ASD: autism spectrum disorder; ASD-diet: ASD children with dietary intake without gluten and casein; ASD-no diet: children with ASD without restricted diet; ASD-total: children with ASD with and without restricted diet, TD: typical development; T: total. Comparison between: * ASD-diet and ASD-no diet, *p *** ASD-total group and TD group.

**Table 3 nutrients-14-02155-t003:** Comparison of dietary intake between nutritional status groups (normal weight vs. excess weight) and neurodevelopment groups (autism spectrum disorder vs. typical development) by two-way ANCOVA.

Dietary Intake	ASD-Total Group		TD Group		All		Pair Comparisons					Main Effects		Interactions
	Normal Weight (*n* = 20)	Excess Weight (OW + OB) (*n* = 11)	Normal Weight (*n* = 20)	Excess Weight (OW + OB) (*n* = 9)	Normal Weight (*n* = 41)	Excess Weight (OW + OB) (*n* = 20)	*p* ^1^	*p* ^2^	*p* ^3^	*p* ^4^	*p* ^5^	Nutritional Status Factor	Neurodevelopment Factor	Nutritional Status × Neurodevelopment
*Dairy products*														
Milk + Yogurt (g/day)	81.23 ± 189.59	106.59 ± 231.30	400.69 ± 244.35	248.33 ± 193.17	237.06 ± 269.20	170.37 ± 221.59	0.848	0.224	**0.007**	0.243	0.614	NS	**	NS
Cheese (g/day)	5.02 ± 10.11	12.90 ± 14.76	17.64 ± 15.21	14.30 ± 16.54	11.17 ± 14.21	13.53 ± 15.18	0.901	0.140	0.079	0.831	0.400	NS	NS	NS
T. Dairy (g/day)	120.10 ± 229.46	160.49 ± 253.98	453.36 ± 248.11	287.70 ± 202.04	282.67 ± 289.84	217.73 ± 235.27	0.588	0.102	**0.014**	0.697	0.530	NS	NS	NS
*Vegetable drinks*														
Vegetable Drinks (ml/day)	295.85 ± 403.33	89.45 ± 204.80	38.75 ± 141.99	0.00 ± 0.00	170.43 ± 328.39	49.20 ± 155.43	0.266	-	0.382	-	0.506	NS	NS	-
*Cereals*														
Cereals with gluten (g/day)	79.66 ± 127.14	142.71 ± 145.27	246.18 ± 104.89	215.25 ± 116.99	160.89 ± 142.86	175.35 ± 135.06	0.245	0.544	0.079	0.980	0.688	NS	NS	NS
Cereals without gluten (g/day)	107.07 ± 96.66	77.23 ± 84.41	18.94 ± 24.44	54.88 ± 96.75	64.08 ± 83.33	67.17 ± 88.44	0.274	0.271	0.131	0.525	0.725	NS	NS	NS
T. Cereals (with and without gluten) (g/day)	186.73 ±108.84	219.95 ± 127.93	265.13 ± 105.57	270.13 ± 109.65	224.97 ± 113.09	242.53 ± 119.72	0.697	0.564	0.289	0.115	0.898	NS	NS	NS
*Meat, fish, and eggs*														
Meat (g/day)	37.07 ± 25.47	44.45 ± 29.81	33.80 ± 20.13	26.88 ± 12.65	35.47 ± 22.79	36.55 ± 24.81	0.202	0.676	0.465	0.455	0.537	NS	NS	NS
Minced meat (g/day)	21.13 ± 24.79	70.07 ± 144.59	20.54 ± 15.11	17.17 ± 13.85	20.84 ± 20.39	46.27 ± 108.69	0.250	0.683	**0.010**	0.994	0.185	NS	NS	NS
Chicken (g/day)	43.47 ± 52.78	38.27 ± 24.40	29.10 ± 21.50	30.44 ± 14.51	36.46 ± 40.81	34.75 ± 20.44	0.956	0.722	0.474	0.736	0.775	NS	NS	NS
Pork (g/day)	12.23 ± 20.81	4.16 ± 8.08	2.60 ± 3.32	2.51 ± 4.36	7.53 ± 15.67	3.42 ± 6.56	0.685	0.696	0.527	0.627	0.576	NS	NS	NS
*Milanesa* with gluten (g/day)	15.72 ± 40.73	12.76 ± 10.56	13.37 ± 9.70	22.16 ± 16.86	14.57 ± 29.59	16.99 ± 14.19	0.891	0.175	0.798	0.457	0.296	NS	NS	NS
*Milanesa* without gluten (g/day)	6.17 ± 10.02	5.45 ± 18.09	1.80 ± 5.16	0.00 ± 0.00	4.04 ± 8.23	3.00 ± 13.41	0.067	.	0.557	.	0.152	NS	NS	NS
Eggs (g/day)	46.85 ± 66.02	21.60 ± 22.18	29.49 ± 19.88	22.66 ± 16.13	38.38 ± 49.44	22.08 ± 19.20	0.395	0.504	0.530	0.462	0.285	NS	NS	NS
*Fish*														
Fresh fish (g/day)	0.04 ± 0.17	0.00 ± 0.00	7.00 ± 5.13	10.95 ± 12.19	3.43 ± 4.99	4.93 ± 9.69	-	0.942	-	-	-	NS	-	-
Canned fish (g/day)	0.00 ± 0.00	0.00 ± 0.00	7.76 ± 11.15	1.24 ± 3.73	3.78 ± 8.63	0.56 ± 2.50	-	0.809	-	-	0.809	NS	-	-
Fish (fresh and canned) (g/day)	0.04 ± 0.17	0.00 ± 0.00	14.76 ± 11.27	12.20 ± 15.62	7.22 ± 10.76	5.49 ± 11.89	-	0.129	.	.	.	NS	.	.
T. foods source of proteins (g/day)	193.77 ± 95.40	210.89 ± 150.39	163.09 ± 51.59	155.85 ± 51.31	178.80 ± 77.82	186.12 ± 117.48	0.815	0.667	0.339	0.542	0.641	NS	NS	NS
*Butter, ghee, and oils*														
Butter (g/day)	0.29 ± 0.63	2.52 ± 7.47	3.55 ± 6.06	1.55 ± 3.21	1.88 ± 4.51	2.09 ± 5.83	0.590	0.104	0.306	0.517	0.847	NS	NS	NS
Ghee (g/day)	1.62 ± 5.44	0.00 ± 0.00	0.08 ± 0.31	0.00 ± 0.00	0.87 ± 3.93	0.00 ± 0.00	.	.	.	.	.	NS	NS	NS
Coconut oil (g/day)	7.79 ± 11.22	1.64 ± 3.87	0.03 ± 0.06	3.00 ± 6.00	4.00 ± 8.85	2.25 ± 4.85	0.718	**0.003**	**0.011**	0.552	**0.026**	*	NS	***
Oil (g/day)	23.23 ± 17.64	23.18 ± 14.96	20.65 ± 14.96	19.32 ± 9.00	21.97 ± 16.23	21.44 ± 12.48	0.648	0.927	0.633	0.961	0.796	NS	NS	NS
T. foods source of fat (g/day)	34.13 ± 23.76	27.35 ± 17.03	24.32 ± 16.94	23.87 ± 12.92	29.35 ± 21.05	25.79 ± 15.04	0.924	0.979	0.627	0.773	0.961	NS	NS	NS

Results expressed as means ± SD. Statistically significant differences (indicated in bold, *p* < 0.05; adjusted for birth weight, sensory profile, with/without diet): (a) Two-way ANCOVA, comparisons between: ^1^ normal and excess weight in ASD group, ^2^ normal and excess weight in TD group, ^3^ normal weight in ASD and TD, ^4^ excess weight in ASD and TD; * significant main effect of nutritional status group factor, ** significant main effect of neurodevelopment factor, *** significant interactions between nutritional status groups and measure moment (significant main effects and interactions are described in result section), NS: no significant; (b) One-way ANCOVA, comparison between ^5^ normal weight and excess weight in all participants. ASD: autism spectrum disorder; ASD-total group: children with ASD with and without restricted diet, TD: typical development, T: total, OW: overweight, OB: obesity, All: ASD + TD children.

**Table 4 nutrients-14-02155-t004:** Comparison of dietary intake between sensorial profile groups (typical performance and probable + definite difference) and neurodevelopment groups (autism spectrum disorder vs. typical development group) by two-way ANCOVA.

Dietary Intake	ASD-Total Group		TD Group		All		Pair Comparisons					Main Effects		Interactions
	Typical Performance (*n* = 11)	Probable + Definite Difference (*n* = 21)	Typical Performance (*n* = 27)	Probable + Definite Difference (*n* = 2)	Typical Performance (*n* = 38)	Probable + Definite Difference (*n* = 23)	*p* ^1^	*p* ^2^	*p* ^3^	*p* ^4^	*p* ^5^	Sensory Profile Factor	Neurodevelopment Factor	SP × ND Factor
*Dairy products*														
Milk + Yogurt (g/day)	95.18 ± 186.15	87.21 ± 213.50	358.28 ± 244.94	287.50 ± 88.38	282.12 ± 257.19	104.62 ± 212.42	**0.009**	0.958	**0.004**	0.640	0.085	NS	NS	NS
Cheese (g/day)	8.91 ± 12.37	7.10 ± 12.48	14.36 ± 13.43	46.80 ± 2.54	12.78 ± 13.21	10.56 ± 16.51	0.064	0.051	0.274	0.292	**0.009**	*	NS	NS
T. Dairy (g/day)	8.91 ± 12.37	7.10 ± 12.48	14.36 ± 13.43	46.80 ± 2.54	12.78 ± 13.21	10.56 ± 16.51	**0.019**	0.532	**0.037**	0.950	0.061	NS	NS	NS
*Vegetable drinks*														
Vegetable drinks (ml/day)	206.54 ± 378.24	234.52 ± 356.66	28.70 ± 122.61	0.00 ± 0.00	80.18 ± 236.45	214.13 ± 346.71	0.367	^.^	0.365	.	0.228	NS	NS	NS
*Cereals*														
Cereals with gluten (g/day)	104.24 ± 105.77	99.81 ± 150.17	237.04 ± 111.57	230.40 ± 27.44	198.60 ± 124.48	111.17 ± 148.16	0.148	0.962	0.152	0.991	0.458	NS	NS	NS
Cereals without gluten (g/day)	69.06 ± 71.86	111.35 ± 100.00	30.84 ± 60.14	20.00 ± 0.00	41.90 ± 65.16	103.40 ± 98.91	0.289	0.591	0.755	0.152	0.936	NS	NS	NS
T. Cereals (with and without gluten) (g/day)	130.15 ± 97.38	173.25 ± 117.86	235.17 ± 103.72	216.43 ± 6.54	204.77 ± 111.59	177.00 ± 113.07	**0.036**	0.799	0.060	0.672	0.419	NS	NS	NS
*Meats, fish, and eggs*														
Meat (g/day)	45.36 ± 29.14	36.60 ± 25.70	31.07 ± 18.75	39.50 ± 0.70	35.21 ± 22.80	36.85 ± 24.52	**0.047**	0.632	0.912	0.153	0.578	NS	NS	NS
Minced meat (g/day)	14.74 ± 16.79	50.11 ± 106.49	17.97 ± 13.92	40.00 ± 0.00	17.04 ± 14.65	49.23 ± 101.57	0.247	0.268	0.203	0.291	0.130	NS	NS	NS
Chicken (g/day)	26.54 ± 13.50	49.61 ± 53.01	29.48 ± 19.88	30.00 ± 14.14	28.63 ± 18.14	47.91 ± 50.95	0.137	0.739	0.594	0.469	0.313	NS	NS	NS
Pork (g/day)	7.03 ± 12.09	10.73 ± 20.28	2.62 ± 3.71	1.90 ± 1.55	3.90 ± 7.30	9.96 ± 19.50	0.736	0.899	0.550	0.998	0.959	NS	NS	NS
*Milanesa* with gluten (g/day)	11.12 ± 10.91	16.58 ± 40.59	16.10 ± 12.64	16.12 ± 19.62	14.66 ± 12.24	16.54 ± 38.93	0.179	0.586	0.632	0.330	0.859	NS	NS	NS
*Milanesa* without gluten (g/day)	1.70 ± 2.37	8.14 ± 15.70	1.33 ± 4.48	0.00 ± 0.00	1.44 ± 3.96	7.43 ± 15.15	0.118	.	0.517	.	0.342	NS	NS	NS
Eggs (g/day)	27.81 ± 34.52	42.60 ± 64.41	28.08 ± 19.23	17.79 ± 8.06	28.01 ± 24.12	41.34 ± 61.89	0.374	0.762	0.282	0.817	0.849	NS	NS	NS
*Fish*														
Fresh fish (g/day)	0.00 ± 0.00	0.03 ± 0.17	7.133 ± 5.33	23.00 ± 24.04	5.06 ± 5.54	2.03 ± 8.37	.	0.168	.	.	.	NS	.	.
Canned fish (g/day)	0.00 ± 0.00	0.00 ± 0.00	5.33 ± 10.14	11.20 ± 0.00	0.97 ± 3.22	2.72 ± 7.34	.	0.892	.	.	0.892	NS	.	.
Fish (fresh and canned) (g/day)	0.00 ± 0.00	0.04 ± 0.17	12.47 ± 10.65	34.20 ± 24.04	8.86 ± 10.60	3.01 ± 11.09	.	0.085	.	.	.	NS	.	.
T. food source of proteins (g/day)	147.08 ± 62.82	227.20 ± 127.38	158.73 ± 50.84	189.31 ± 54.88	155.36 ± 53.96	223.90 ± 122.50	**0.034**	0.515	0.423	0.905	0.112	NS	NS	NS
*Butter, ghee, and oils*														
Butter (g/day)	0.43 ± 0.76	1.39 ± 5.43	2.86 ± 5.45	3.90 ± 5.51	2.16 ± 4.72	1.60 ± 5.36	0.674	0.388	0.637	0.525	0.335	NS	NS	NS
Ghee (g/day)	0.29 ± 0.67	1.47 ± 5.46	0.06 ±0.27	0.00 ± 0.00	0.13 ± 0.43	1.34 ± 5.22	0.605	.	.	.	.	NS	.	.
Coconut oil (g/day)	5.43 ± 11.54	5.80 ± 8.96	1.02 ± 3.62	0.00 ± 0.00	2.30 ± 7.02	5.30 ± 8.70	0.436	.	**0.049**	.	0.650	NS	**	NS
Oil (g/day)	22.08 ± 16.19	23.81 ± 17.05	18.95 ± 12.34	37.50 ± 17.67	19.86 ± 13.41	25.00 ± 17.15	0.576	0.150	0.984	0.283	0.129	NS	NS	NS
T. food source of fat (g/day)	28.24 ± 24.25	33.67 ± 20.53	22.91 ± 14.70	41.40 ± 23.19	24.45 ± 17.79	34.34 ± 20.31	0.745	0.189	0.858	0.250	0.194	NS	NS	NS

Results expressed as means ± SD. Statistically significant differences (indicated in bold, *p* < 0.05; adjusted for birth weight, nutritional status, with/without diet): (a) Two-way ANCOVA, comparisons between: ^1^ typical performance and probable + definite deference in ASD group, ^2^ typical performance and probable + definite deference in TD group, ^3^ typical performance in ASD and TD, ^4^ probable + definite deference in ASD and TD, * significant main effect of sensory profile group factor, ** significant main effect of neurodevelopment factor (significant main effects and interactions are described in result section), NS: no significant; (b) One-way ANCOVA, comparison between ^5^ typical performance and probable + definite deference in all participants. ASD: autism spectrum disorder; ASD-total group: children with ASD with and without restricted diet, TD: typical development, T: total, SP: sensory profile, ND: neurodevelopment, All: ASD + TD children.

## Data Availability

Not applicable.

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
