# Peer review of "Dietary Intake, Nutritional Status and Sensory Profile in Children with Autism Spectrum Disorder and Typical Development"

_nutrients, 2022, doi:10.3390/nu14102155_

Round 1

Reviewer 1 Report

The manuscript entitled „Dietary intake, nutritional status and sensory profile in children with autism spectrum disorder and typical development” describes a study on the nutritional status of children with ASD.

Although the group was small, I believe that this manuscript is an important contribution, since the frequency of ASD is growing and there is still a lot to discover.

I have some small comments worth considering to improve the overall quality of the article.

Please, use the person-first language throughout the text for both ASD and TD. “ASD children” is not a correct term.

Please put the mane of the project in Spanish into “ “ or in italic, to highlight that it is the title.

Line 146: why products without gluten and rice are together? Rice is the base of the gluten-free diet

Please correct the formatting of subsections according to MDPI requirements.

From line 202: there is no need to run a study to guess that children on a gluten-free and casein-free diet will consume fewer cereals and dairy. It can be mentioned, but not as the main finding.

Please do not repeat values from the Tables again in the text.

Author Response

Response to Reviewer 1 Comments

We acknowledge your suggestions and comments to improve our work. We have considered all the suggestions and have incorporated them into the revised manuscript.

Point 1: Please, use the person-first language throughout the text for both ASD and TD. “ASD children” is not a correct term.

Response 1: We substitute “ADS children” for the term “children with ASD” taking into account your observation. Thank you.

Point 2: Please put the name of the project in Spanish into “ “ or in italic, to highlight that it is the title.

Response 2: According to your observation, we modified and put the name of the project in italic.

Point 3: Line 146: why products without gluten and rice are together? Rice is the base of the gluten-free diet.

Response 3: We wanted to study the consumption of gluten-free cereals and for this reason we included products made with gluten-free flours such as pasta, bread, cookies, bakery products, breakfast cereals, pizza, and empanadas, and we added rice because it is a cereal also used in this diet in its natural form.  Regardless, if there are any further questions regarding grouping we can expand.

Point 4: Please correct the formatting of subsections according to MDPI requirements.

Response 4:  Formatting has now been corrected as per comment. Thank you for the observation.

Point 5: From line 202: there is no need to run a study to guess that children on a gluten-free and casein-free diet will consume fewer cereals and dairy. It can be mentioned, but not as the main finding.

Response 5: Regarding this comment, we agree and modify it by adding at the beginning of the line “As expected”, to avoid making it sound like a finding when in fact, as you stated, these are expected results. It should be noted that we mentioned that the ASD-diet group had a lower intake of total dairy products, cereals with gluten, total cereals, milanesa with gluten and butter than the ASD-no diet because  the difference between ASD-total and TD groups may be due to the ASD-diet group.

Point 6: Please do not repeat values from the Tables again in the text.

Response 6: We agree and modified as per comment, values were removed from text and only left the p value in it.

Reviewer 2 Report

In the manuscript titled '" Dietary intake, nutritional status and sensory profile in children with autism spectrum disorder and typical development" by Mendive and Guerendiain, the authors carried out a study in Montevideo, Uruguay, involving 65 children aged 3 to 12 years with and without ASD in order to investigate the dietary intake, nutritional status and sensory profile with ASD. The Short Sensory Profile and food frequency questionnaires revealed that children with ASD had a higher intake (gr or ml/d) of vegetable drinks (p=0.001), gluten-free cereals (p=0.003), and a lower intake of fish (p<0.001) than TD ones, as well as a lower score in total sensory profile score. This work appears to be worth publishing in Nutrients. However, a few minor issues need to be addressed and clarified.

  • Please check the typographical and grammatical errors throughout the manuscript.
  • The presented Tables should be better described.

Author Response

Response to Reviewer 2 Comments

We acknowledge your suggestions and comments to improve our work.

Point 1: “Please check the typographical and grammatical errors throughout the manuscript”.

Response 1: We apologize for the typographical and grammatical errors. The manuscript has been reviewed and corrected.

Point 2: “The presented Tables should be better described”.

Response 2: Table description has been improved as per recommendation, but we are open and grateful for any additional specifics.
